# Trends in gestational age at birth in the city of São Paulo, Brazil between 2012 and 2019

**Margarida Maria Tenório de Azevedo Lira**[1�], **Marina de Freitas**[1�],
**Eliana de Aquino Bonilha**[2�], **Célia Maria Castex Aly**[1�], **Patrícia Carla dos Santos**[1],
**Denise Yoshie Niy**[ID][3�*], **Carmen Simone Grilo Diniz**[4�]

**1** Potential Pregnancy Days Lost Study, São Paulo, Brazil, **2** São Camilo University Center, São Paulo, Brazil, **3** Gender, Evidence and Health Study Group, School of Public Health, University of São Paulo, São Paulo, Brazil, **4** Department of Health and Life Cycles, School of Public Health, University of São Paulo, São Paulo, Brazil

These authors contributed equally to this work.
\* denise.niy@gmail.com

## Abstract

Studies have shown that excessive obstetric interventions such as induced labor and caesarean sections have contributed to the shortening of the length of gestation, leading to a left shift in gestational age (GA) at birth. The aim of this study was to analyze trends in GA and the contribution of associated factors to changes in São Paulo city, Brazil during the period 2012–2019. We conducted an observational time-series study of births in São Paulo using data from Brazil's national live births information system (SINASC). We calculated the annual percent change (APC) of births by GA between 2012 and 2019 and between the first and second four-year periods of the time series by applying log transformation to the percentages, followed by Prais-Winsten regression. A total of 1,525,759 live births were analyzed. From 2015, there was an increase in the proportion of live births between 39 and 40 weeks from 2015 and a fall in the proportion of early term (37–38 weeks) and preterm (< 37 weeks) births throughout the study period. The APC of births at 39 and 40 weeks was 7.9% and 5.7%, respectively, while the proportion of births at other gestational ages showed a statistically significant reduction over the study period. These reductions were more pronounced in the first four-year period (2012–2015). The same trend was observed when the data were analyzed by type of delivery, type of service (public or private), maternal age, and maternal education level. The findings show that there was a right shift in the GA curve during the study period and a reduction in the proportion of preterm and early term births. These changes were more pronounced in births that occurred in private hospitals. These changes reflect public policies implemented to reduce obstetric interventions such as induced labor and caesarean section before labor, especially before 39 weeks of gestation.

**Data availability statement:** The database is fully available for consultation at the Harvard Dataverse Repository: https://doi.org/10.7910/DVN/PP2VVF.

**Funding:** The project was developed in partnership with the São Paulo City Department of Health, Bill & Melinda Gates Foundation (reference number OPP1201939 - https://www.gatesfoundation.org/), and National Council for Scientific and Technological Development (reference number 443775/2018 – Fundação Bill e Melinda Gates/CNPq/DECIT-MS - https://www.gov.br/cnpq/pt-br). CSGD is a Research Productivity Fellow - Level 2 from the National Council for Scientific and Technological Development. MMTAL received a Industrial Technological Development Scholarship - Level B from the National Council for Scientific and Technological Development. MF received a Industrial Technological Development Scholarship - Level B from the National Council for Scientific and Technological Development. The funders had no role in study design, data collection and analysis, decision to publish, or preparation of the manuscript.

**Competing interests:** The authors have declared that no competing interests exist.

## Introduction

Length of gestation is one of the leading predictors of newborn health outcomes. Gestational age (GA) can be estimated using different methods and is generally measured in completed weeks. Until recently, GA was treated as a binary question, with newborns being considered either preterm (< 37 weeks) or term (37 0/7–41 6/7 weeks), justifying interventions such as labor induction or caesarean section from 37 weeks, when fetal development may not be complete. Studies have shown that the term period should not be treated uniformly, as early term infants (37–38 weeks) tend to have a higher risk of morbidity and mortality than those born between 39 and 41 weeks, often showing similar outcomes to late preterm infants [1–7]. In 2013, the American College of Obstetricians and Gynecologists [8] proposed a different classification of GA, dividing the term period into three categories: early term (37–38 weeks), full term (39–40 weeks), and late term (41 weeks).

The Live Births Information System of Brazil (SINASC) was established in 1996 to systematically collect data on live births across the national territory. Designed to support all levels of Brazil's healthcare system, SINASC has consistently demonstrated high coverage, completeness, and reliability in its recorded variables. This ensures its capacity to fulfill its primary objective: to provide comprehensive and objective analyses of the healthcare landscape, thereby informing policies that enhance maternal and child health.

In São Paulo, efforts to improve data quality have included rigorous monitoring and continuous professional training for those responsible for completing and inputting information into the Certificate of Live Birth (CLB), SINASC's foundational reporting form. The training process encompasses the development of educational materials, seminars, and both individual and group workshops. Additionally, healthcare facilities that conduct births and adhere to established standards of completeness and timely data entry receive annual certification through the "SINASC Seal", reinforcing quality assurance and data integrity [9–11].

SINASC used to record GA as a categorical variable in weekly intervals up to 2010, until the CLB was modified to record this information as a continuous variable. This change enabled a more comprehensive analysis of the distribution of GA. In the city of São Paulo, the new CLB began to be used in 2011, but full implementation across all health facilities only occurred in 2012.

The average physiological length of pregnancy is 280 days, or 40 weeks [12,13]. However, interventions such as scheduled labor induction or caesarean section can shorten the length of gestation [7,14,15]. In Brazil, caesarean section rates have been far higher than the World Health Organization recommended rate of 15% for decades, reaching 57% in the general population [16,17].

In a study investigating the contribution of private sector deliveries in the city of São Paulo to reductions in length of gestation, Diniz et al. [5] found a one-week left shift in GA in infants born by caesarean section through the private health system; the same result was reported by Raspantini et al [18]. A study in Australia [7] investigating trends in the distribution of GA and the contribution of planned births (induced

labor and elective caesarean section) to changes also observed a left shift in GA at birth and the findings suggest a changing pattern towards fewer births commencing spontaneously and increasing planned births.

The aim of this study was to analyze trends in GA at birth and the contribution of associated factors in the city of São Paulo during the period 2012–2019.

## Materials and methods

### Type of study and data source

We conducted an observational time-series study of births that occurred in the city of São Paulo (SP) in the period 2012–2019 using data from the SINASC. The anonymized databases were provided by the São Paulo City Department of Health on May 6, 2020. This study is part of the Potential Pregnancy Days Lost project, which was approved by the Research Ethics Committee of the University of São Paulo's School of Public Health (CAAE: 98163018.2.0000.5421), on October 11, 2018. Since the analysis used secondary data, individual consent was not required.

### Study population and variables

The variables studied were: maternal age and education, mode of delivery, labor induction, type of pregnancy, gestational age, birth weight, type of service, mother's municipality residence.

We studied liveborn delivered in public and private hospitals at 22–45 weeks of gestation, weighing ≥ 500 grams born to mothers aged between 10 and 49 years. A total of 1,525,759 live births were recorded in SP between 2012 and 2019. Of these, 12,382 (0.8%) were excluded because they did not meet the inclusion criteria (mothers aged under 10 or over 49 years or maternal age not recorded, 272; type of pregnancy not recorded, 40; GA at birth under 22 or over 45 weeks or GA not recorded, 4,388; birth weight < 500 grams or weight not recorded, 922; out-of-hospital births or place of birth not recorded, 6,746; and mode of delivery not recorded, 14), resulting in a final sample of 1,513,377 (99.2%) (Table 1). All variables analyzed in this study showed completeness ranging from 98.9% to 100.0%.

The choice to analyze births occurring in SP stemmed from the need to evaluate the quality and effectiveness of childbirth services within the health system. Given its significance for public health planning, this information is particularly valuable for the health system management of São Paulo city, offering insights that can inform policies and improve maternal care.

The results of the exploratory analysis showed that the GA curves for all live births and singleton births overlapped. These types of pregnancy were therefore analyzed together for the purposes of the present study (Fig 1).

### Statistical analysis

First, we performed a descriptive statistical analysis of the data, followed by an analysis of trends in GA at birth by calculating the annual percent change of live births over the period 2012–2019. The results showed an increase in the

**Table 1. Study population and exclusions. City of São Paulo, Brazil, 2012–2019.**

| Exclusion criteria | Exclusions | Study population |
|---|---|---|
| Total | | 1,525,759 |
| Mothers aged under 10 and over 49 years or maternal age not recorded | 272 | 1,525,487 |
| Type of pregnancy not recorded | 40 | 1,525,447 |
| GA at birth under 22 or over 45 weeks or GA not recorded | 4,388 | 1,521,059 |
| Birth weight < 500 grams or weight not recorded | 922 | 1,520,137 |
| Out-of-hospital births or place of birth not recorded | 6,746 | 1,513,391 |
| Mode of delivery not recorded | 14 | 1,513,377 |

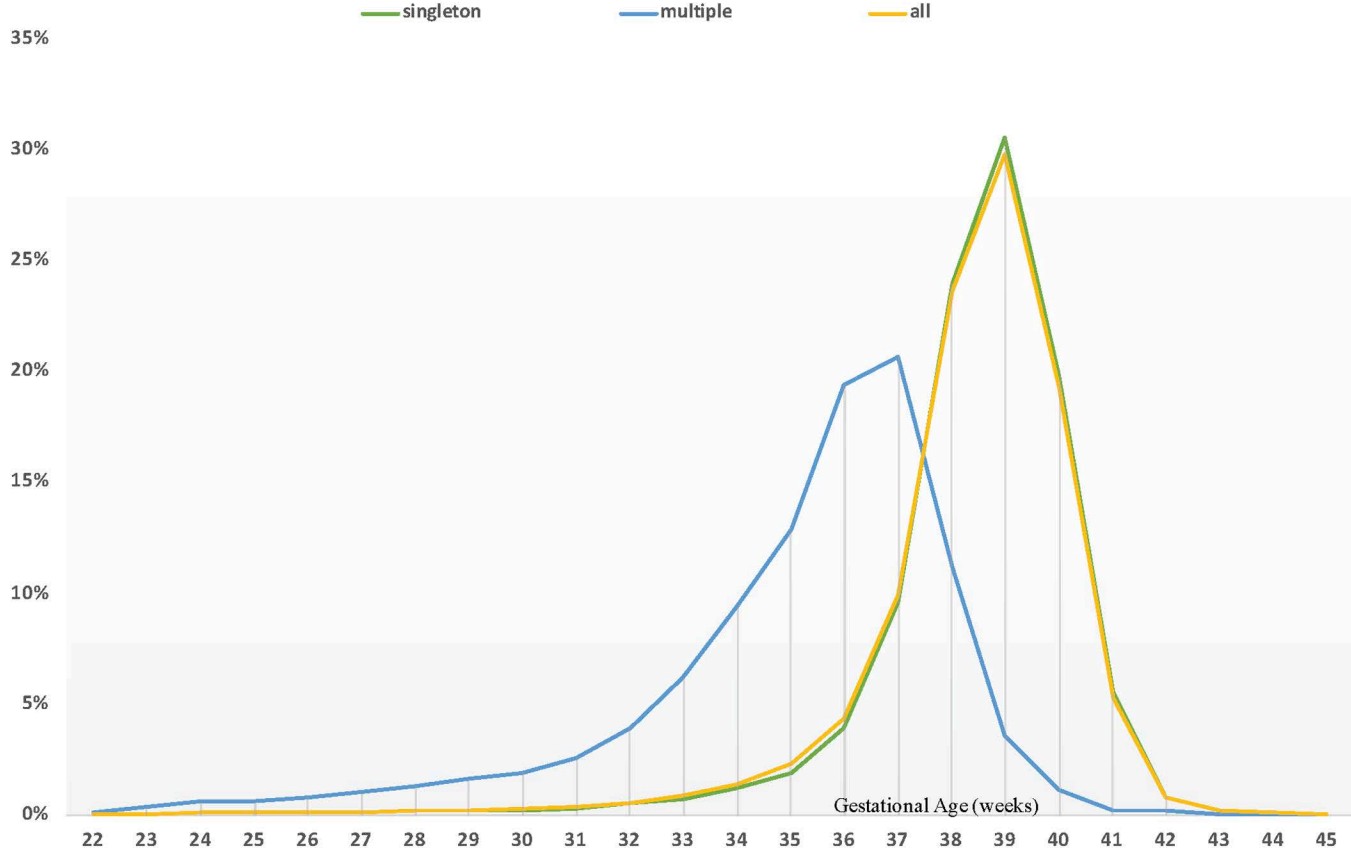

**Fig 1. Live births by type of pregnancy.** City of São Paulo, Brazil, 2012–2019.

proportion of births between 39 and 40 weeks from 2015. Differences in percent change rates were particularly pronounced between private and public sector births and mode of delivery [19]. A t-test for independent proportions was used to assess whether these differences were statistically significant. We therefore opted to compare the first and last four-year periods (2012–2015 and 2016–2019, respectively) (Fig 2).

Percent change rate was calculated by applying log transformation to the percentages, followed by Prais-Winsten regression [20] to estimate annual percent change ($\beta 1$) in births by GA. Subsequently, the $\beta_1$ values obtained were applied to the following formula to calculate the percent variation rates:

$$Rate = [-1 + 10^{\beta 1}] \times 100$$

The confidence intervals (CI) of the percent change rates were calculated based on the maximum and minimum $\beta$ values using the following formula:

$$95\%CI = [-1 + 10\beta minimum \ X \ 100; \ -1 + 10\beta maximum \ x \ 100]$$

The null hypothesis ($H_0$) was that the trend is stationary, i.e., there is no significant difference between the percent change rate and zero, adopting a 95% confidence level. Depending on the percent change rate, the trend may either be increasing (positive value), decreasing (negative value), or stationary, when the null hypothesis is accepted. In the

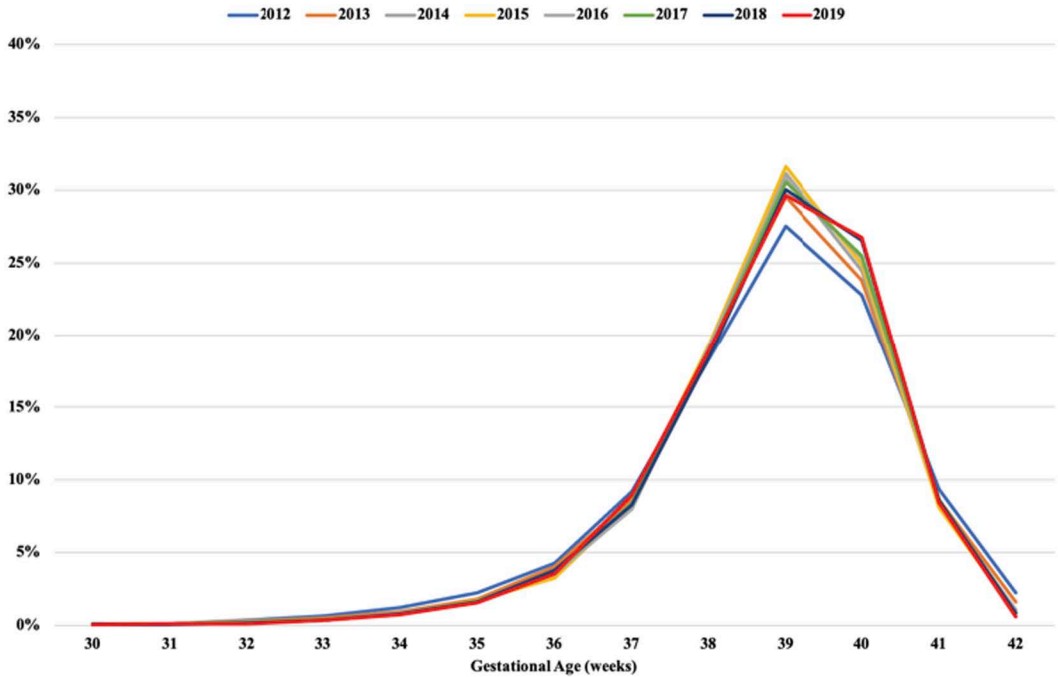

**Fig 2. Distribution of live births by gestational age.** City of São Paulo, Brazil, 2012–2019.

Prais-Winsten regression, the t-test was used to assess the statistical significance of the estimated coefficients. This test evaluates whether each coefficient is significantly different from zero by testing the null hypothesis ($H_0$) against the alternative hypothesis ($H_A$). Data processing and analysis was performed using SPSS version 20 and R.

## Results

Table 2 shows that the annual number of live births fell by 8.1% over the study period. The proportion of teenage mothers decreased over the period, while the percentage of mothers aged 35 years and over, mothers with a higher level of education, and multiple pregnancies increased. Deliveries were predominantly caesarean sections, with rates remaining constantly above 50%, although the proportion of this type of delivery fell slightly over the study period. The proportion of pre-labor caesarean sections was high, with rates reaching almost 70% throughout the time series.

The proportion of induced labor vaginal births fell over the period, from 55.0% in 2012 to 27.7% in 2019. With regard to length of gestation, the rate of full-term births (39–40 weeks) increased from 43.5% in 2012 to 53.0% in 2019, while the proportion of births at other gestational ages decreased. The rate of preterm births fell from 12.1% in 2012 to 10.4% in 2015 and remained relatively stable thereafter (Table 2).

Fig 2 shows that the proportion of live births between 30 and 35 weeks remained stable over the study period, while the proportion of births between 36 and 38 weeks fell from 2015, when the percentage of births between 39 and 40 weeks increased.

Fig 3 shows that the most common GA for both spontaneous and induced labor vaginal births was 39 weeks throughout the period. There was an increase in the proportion of spontaneous vaginal births at 39 and 40 weeks over the study period. The proportion of induced labor vaginal births at 39 weeks increased up to 2015, while the proportion of this type of birth at 40 weeks increased over the whole period at a proportionately higher rate than spontaneous vaginal births.

**Table 2. Annual number of live births and proportions according to maternal sociodemographic and birth characteristics. City of São Paulo, Brazil, 2012–2019.**

| | 2012 | 2013 | 2014 | 2015 | 2016 | 2017 | 2018 | 2019 |
|---|---|---|---|---|---|---|---|---|
| **All Newborn** | 191,827 | 191,549 | 196,780 | 196,813 | 187,624 | 188,574 | 183,829 | 176,381 |
| **Maternal age (years)** | | | | | | | | |
| <20 | 12.5 | 12.4 | 12.2 | 11.7 | 11.2 | 10.4 | 9.6 | 9.0 |
| 20-34 | 70.5 | 69.9 | 69.2 | 68.9 | 68.5 | 67.9 | 67.5 | 67.3 |
| 35 and over | 17.0 | 17.7 | 18.6 | 19.4 | 20.4 | 21.7 | 22.9 | 23.7 |
| **Maternal education** | | | | | | | | |
| Did not complete elementary school | 20.7 | 19.0 | 17.8 | 16.7 | 15.8 | 14.8 | 14.0 | 13.6 |
| Completed high school | 50.6 | 50.7 | 50.3 | 50.5 | 50.7 | 50.7 | 50.7 | 50.9 |
| Completed or undergoing higher education | 28.6 | 30.2 | 31.8 | 32.7 | 33.4 | 34.5 | 35.3 | 35.5 |
| **Mode of delivery** | | | | | | | | |
| Vaginal | 42.5 | 41.2 | 42.0 | 43.7 | 44.0 | 44.7 | 45.8 | 45.5 |
| Cesarean | 57.5 | 58.8 | 58.0 | 56.3 | 56.0 | 55.3 | 54.2 | 54.5 |
| **Labor induction** | | | | | | | | |
| Vaginal | | | | | | | | |
| Yes | 55.1 | 55.0 | 49.1 | 44.6 | 39.7 | 33.9 | 29.9 | 27.7 |
| No | 40.3 | 41.9 | 48.4 | 55.2 | 60.2 | 66.1 | 70.1 | 72.3 |
| NA/ignored | 4.6 | 3.1 | 2.5 | 0.2 | 0.1 | 0.0 | 0.0 | 0.0 |
| Cesarean | | | | | | | | |
| Yes | 13.9 | 12.7 | 12.6 | 12.5 | 12.3 | 11.8 | 11.8 | 12.1 |
| No | 83.4 | 86.0 | 86.2 | 87.2 | 87.5 | 88.2 | 88.2 | 87.9 |
| NA/ignored | 2.7 | 1.3 | 1.2 | 0.2 | 0.2 | 0.0 | 0.0 | 0.0 |
| **CS before labor[a]** | | | | | | | | |
| Yes | 68.3 | 69.9 | 69.7 | 67.5 | 66.7 | 66.9 | 67.8 | 67.5 |
| No | 26.8 | 27.2 | 27.5 | 30.1 | 30.8 | 32.4 | 31.6 | 32.5 |
| NA/ignored | 4.9 | 2.9 | 2.8 | 2.4 | 2.5 | 0.7 | 0.5 | 0.1 |
| **Type of pregnancy** | | | | | | | | |
| Singleton | 97.3 | 97.1 | 97.1 | 97.1 | 97.1 | 97.0 | 97.1 | 97.1 |
| Multiple | 2.7 | 2.9 | 2.9 | 2.9 | 2.9 | 3.0 | 2.9 | 2.9 |
| **Gestational age (weeks)** | | | | | | | | |
| <37 | 12.1 | 11.5 | 10.9 | 10.4 | 10.5 | 10.6 | 10.5 | 10.4 |
| 37–38 | 35.7 | 36.6 | 35.9 | 33.4 | 31.8 | 31.5 | 31.3 | 31.4 |
| 39–40 | 43.5 | 44.5 | 46.4 | 50.1 | 51.5 | 52.1 | 52.8 | 53.0 |
| 41 | 6.3 | 5.8 | 5.6 | 5.2 | 5.2 | 4.9 | 4.7 | 4.6 |
| 42 and over | 2.4 | 1.6 | 1.2 | 0.9 | 1.0 | 0.8 | 0.8 | 0.6 |
| **Birth weight (g)** | | | | | | | | |
| <2,500 | 9.6 | 9.6 | 9.4 | 9.4 | 9.5 | 9.5 | 9.5 | 9.6 |
| 2,500−2,999 | 24.8 | 24.9 | 24.7 | 24.1 | 23.9 | 23.5 | 23.3 | 23.5 |
| 3,000-3,999 | 61.6 | 61.6 | 62.0 | 62.4 | 62.5 | 62.7 | 63.0 | 62.8 |
| ≥4,000 | 3.9 | 3.9 | 3.9 | 4.1 | 4.1 | 4.3 | 4.2 | 4.2 |
| **Type of service** | | | | | | | | |
| Public | 54.6 | 53.7 | 54.0 | 54.4 | 55.7 | 56.6 | 57.4 | 58.2 |
| Private | 45.4 | 46.3 | 46.0 | 45.6 | 44.3 | 43.4 | 42.6 | 41.8 |
| **Mother's municipality of residence** | | | | | | | | |
| São Paulo | 86.6 | 86.2 | 85.2 | 85.5 | 85.3 | 85.9 | 85.9 | 85.8 |
| Other | 13.4 | 13.8 | 14.8 | 14.5 | 14.7 | 14.1 | 14.1 | 14.2 |

[a]Percentages calculated only for CS.

Percentages of labor induced births were calculated separately based on total live births for each type of delivery.

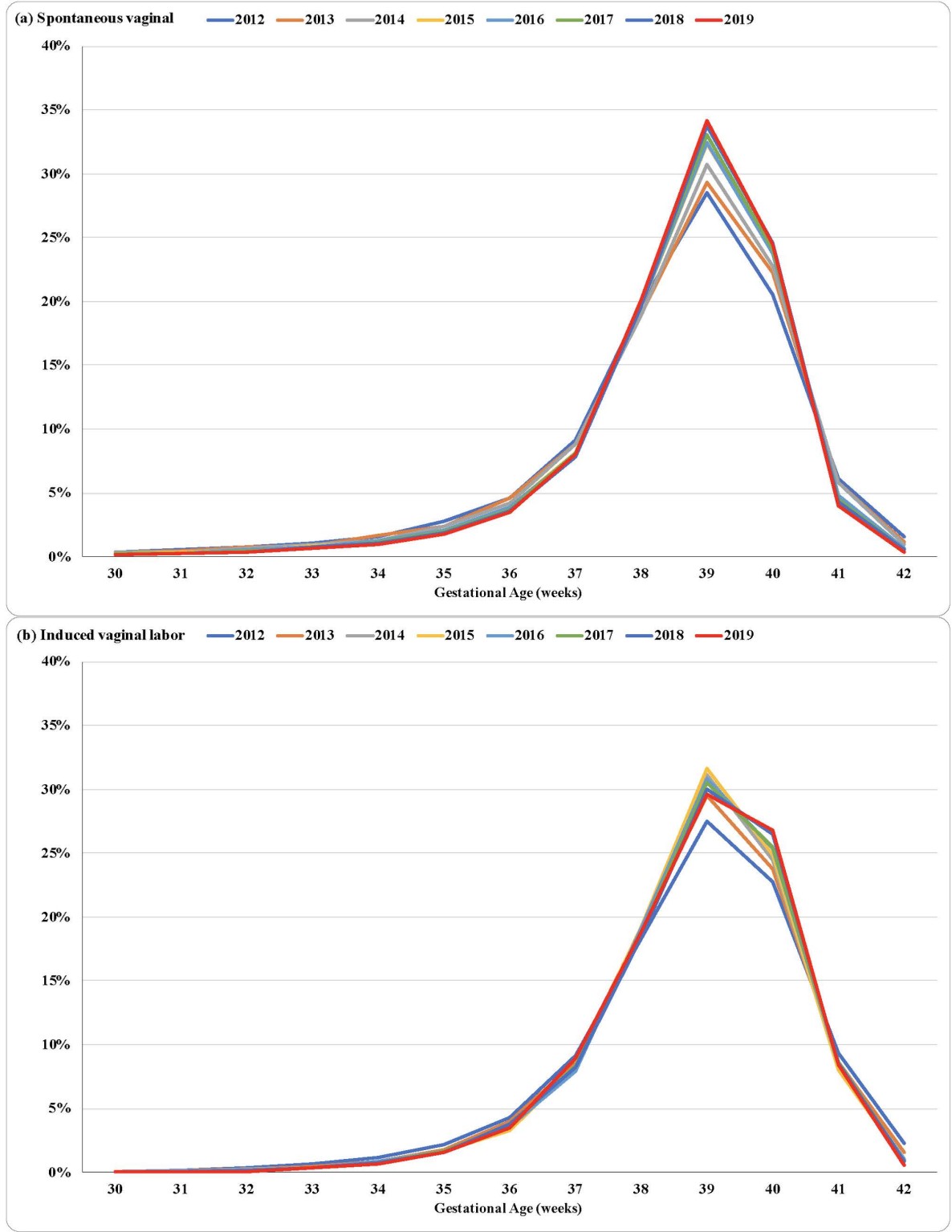

**Fig 3. Distribution of spontaneous (a) and induced labor (b) vaginal births by gestational age.** City of São Paulo, Brazil, 2012–2019.

There was an increase in the proportion of intrapartum caesarean sections at 39 and 40 weeks up to 2015, with rates levelling off in the second four-year period, especially for births at 40 weeks. The most common gestational age for pre-labor caesarean sections was 38 weeks up to 2015, shifting to 39 weeks between 2016 and 2019, with rates remaining above 30% (Fig 4).

There was an increase in the proportion of births at 39 and 40 weeks in public hospitals and normal birth centers over the study period, while in private facilities there was a reduction in the proportion of births at 38 weeks from 2015 and increase in the percentage of births at 39 weeks (Fig 5).

The annual percent change increase in the proportion of births at 39 and 40 weeks was 7.9% and 5.7%, respectively, in the period from 2012 to 2019, while the proportion of births at other gestational ages showed a statistically significant reduction over the study period. These reductions were more pronounced in the first four-year period (2012–2015), especially in 36, 37 and 42 weeks and over. There was also a substantial decline in the proportion of births at less than 37 weeks over the study period. This reduction was also more pronounced in the period 2012–2015 (Table 3). The annual percentage change of gestational age of 42 weeks and over was significantly greater than the others in all periods, with an increase during the period from 2012 to 2015 (52.8%) and a decrease during the period from 2016 to 2019 (26.4%).

Table 4 shows that there was a positive annual percent change in the proportion of vaginal births at 38–40 weeks over the study period. For births at 37 weeks, the annual percent change over the study period was negative (−7.3%) for caesarean births and positive for vaginal births (+1.2%). In the period 2012–2015, the annual percent change in the proportion of caesarean births at 37 weeks was −12.1%, compared to −6.9% for vaginal births. This difference was statistically significant. The proportion of preterm births decreased across all gestational ages under 37 weeks. This reduction was greater in vaginal births. The decline was significantly greater for 42 weeks and over during the period from 2012 to 2015 for both vaginal and cesarean deliveries (58.7% vs. 46.0%).

Table 5 shows that there was a positive annual percent change in the proportion of births at 39 and 40 weeks throughout the study period for births in both public and private hospitals. This change was more pronounced in the period 2012–2019 and for births in private facilities. The proportion of births at 37 weeks decreased over the period 2012–2015 in both public and private hospitals (−5.4% versus 14.1%), while the proportion of births at 39 weeks increased (+11.9% versus +13.5%). The findings also show that there was a reduction in the proportion of late preterm births in both public and private hospitals (Table 5).

Tables 6 and 7 shows that there was a significant positive annual percent change in the proportion of births at 39 and 40 weeks across all age groups and levels of education between 2012 and 2019. During these weeks of gestation, higher rates were observed among mothers aged 35 years and over who had completed more than 12 years of education, compared to other categories of educational level and maternal age. Regarding maternal age, a greater decline was noted between 41 and 42 or more weeks of gestation from 2012 to 2015, compared to the period from 2016 to 2019. Additionally, there was a higher increase in gestational weeks 39 and 40 from 2012 to 2015 compared to 2016–2019 (Table 6).

In weeks 39 and 40, there was a greater decline in the annual percent change during the period from 2012 to 2015 compared to the period from 2016 to 2019 and levels of education. Additionally, APC declined more sharply in gestational weeks 41 and 42 and over, across all maternal education levels and study periods (Table 7).

## Discussion

Our findings show that there was a decrease in the proportion of preterm (< 37 weeks) and early term (37–38 weeks) births between 2012 and 2019 and an increase in the proportion of births at 39 and 40 weeks from 2015. The caesarean section rate declined over the study period. However, this method remained the most common mode of delivery, with rates being higher in private hospitals. The increase in the proportion of births at 39 and 40 weeks was higher for births by caesarean section.

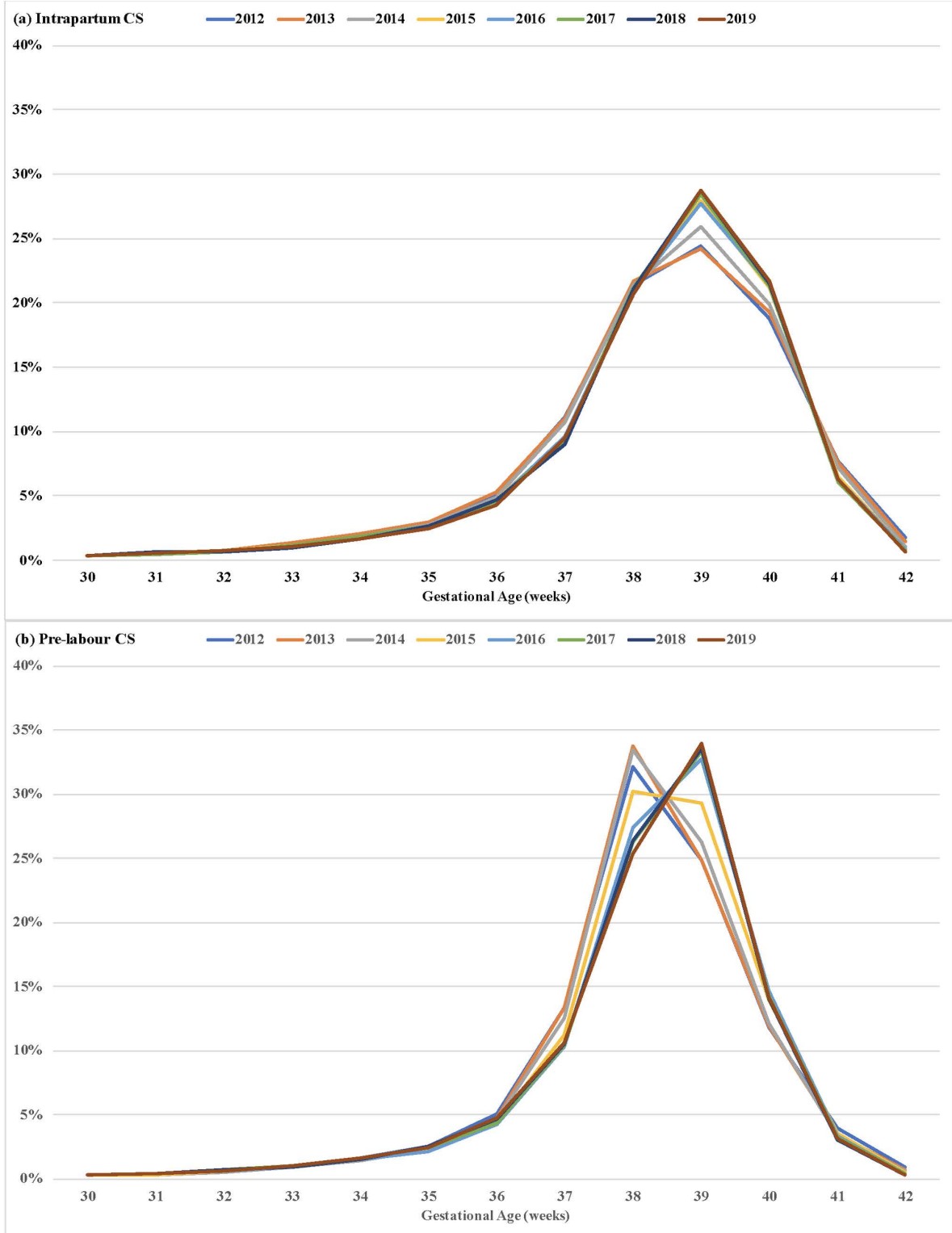

**Fig 4. Distribution of intrapartum (a) and pre-labor (b) caesarean section births by gestational age.** City of São Paulo, Brazil, 2012–2019.

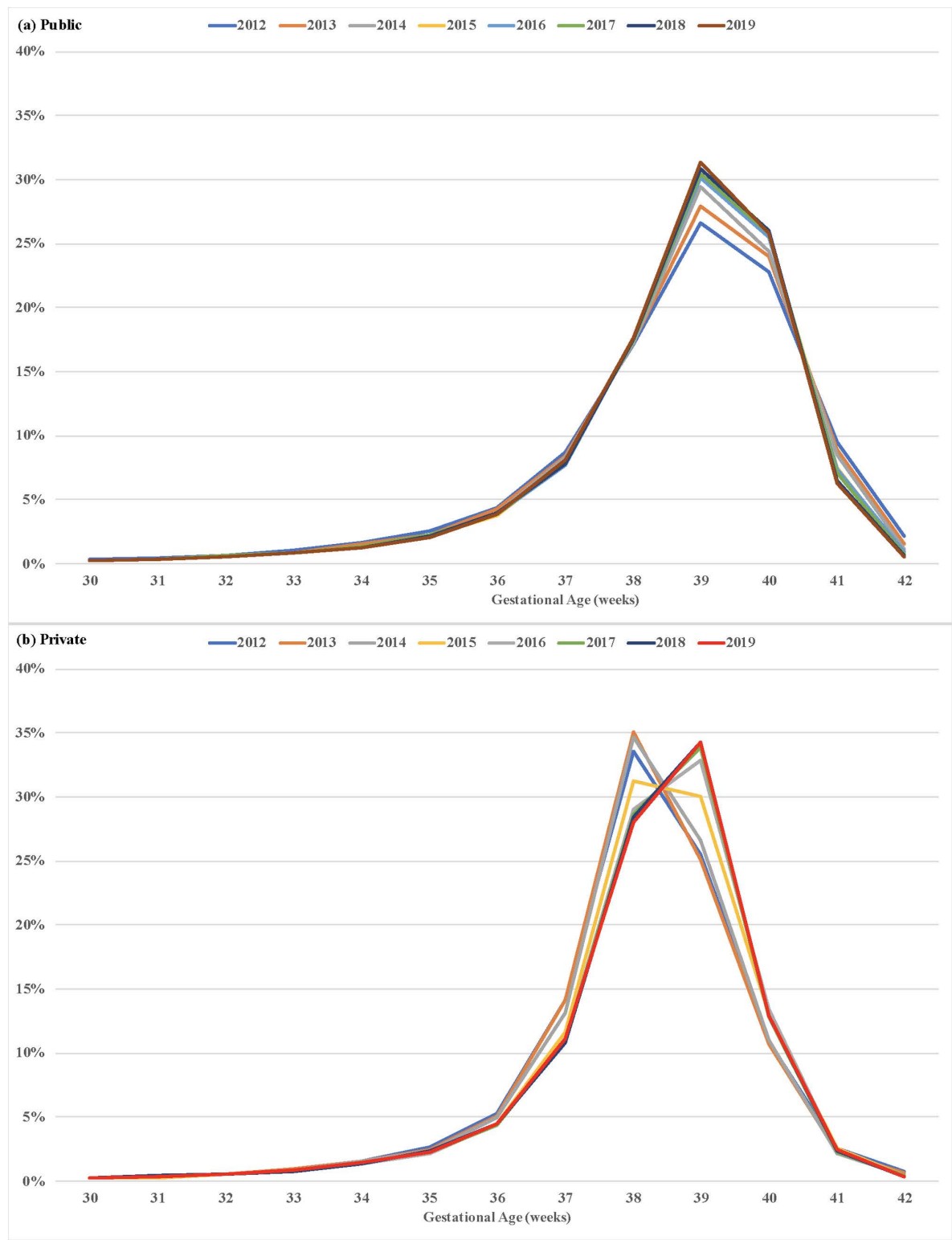

**Fig 5. Distribution of live births in public (a) and private (b) hospitals by gestational age.** City of São Paulo, Brazil, 2012–2019.

**Table 3. Annual percent change of live births by gestational age. City of São Paulo, Brazil, 2012–2019.**

| GA | 2012–2015 | | | 2016–2019 | | | 2012–2019 | | |
|---|---|---|---|---|---|---|---|---|---|
| | rate (%) | | 95% CI | rate (%) | | 95% CI | rate (%) | | 95% CI |
| < 34 | −9.2 | * | (−13.1;-5.1) | −2.7 | * | (−2.7;-2.7) | −3.4 | * | (−6.0;-0.7) |
| 34 | −3.4 | | (−26.3;26.7) | −5.4 | * | (−5.4;-5.4) | −4.3 | * | (−8.4;0.0) |
| 35 | −11.1 | * | (−17.4;-4.3) | −2.1 | | (−11.0;7.7) | −5.4 | * | (−8.9;-1.7) |
| 36 | −10.7 | * | (−15.1;-6.0) | 1.6 | * | (1.6;1.6) | −4.7 | * | (−8.3;-1.0) |
| 37 | −10.7 | * | (−17.6;-3.2) | −0.2 | * | (−0.2;-0.2) | −6.2 | * | (−10.7;-1.5) |
| 38 | −2.5 | | (−11.4;7.2) | −2.1 | * | (−2.8;-1.3) | −4.9 | * | (−7.5;-2.3) |
| 39 | 12.2 | * | (6.6;18.1) | 2.8 | * | (2.1;3.6) | 7.9 | * | (4.4;11.5) |
| 40 | 6.4 | * | (6.4;6.4) | 1.6 | * | (1.6;1.6) | 5.7 | * | (2.8;8.6) |
| 41 | −12.7 | * | (−14.6;-10.8) | −8.4 | * | (−10.4;-6.3) | −9.2 | * | (−10.7;-7.7) |
| 42 and over | −52.8 | * | (−55.2;-50.3) | −26.2 | * | (−26.2;-26.2) | −33.6 | * | (−43.0;-22.7) |

*$p < 0.05$ rejects the null hypothesis.

The results also show that there was a reduction in the proportion of labor induced births over the study period. This may be partially explained by the modification to the CLB in 2011. Data quality has improved over time, especially in the city of São Paulo, where health professionals have received training on how to complete the form and the importance of differentiating between induced and natural labor [21].

International studies have reported a left shift in GA at birth [1,7] and increase in rates of late preterm and early term births related to a rise in obstetric interventions. Caesarean section rates vary widely across countries. However, countries with optimal maternal and neonatal indicators tend to have lower c-section rates, such as Finland (16.0%), Norway (16.1%), the Netherlands (17.3%) and Sweden (17.4%) [22].

Studies in Brazil [6,16,23], including São Paulo [5,18], have shown that scheduled caesarean sections, particularly in private hospitals, and induced labor early term births have contributed to shortening the length of gestation and a left shift in GA at birth. In contrast, the present study observed a right shift in the GA curve from 2015, with an increase in the proportion of births at 39 and 40 weeks and reduction in preterm and early term births. Our results also show that these changes were more pronounced for births in private hospitals. This trend may be partially attributed to a decline in obstetric interventions – such as labor induction and elective caesarean sections – particularly before 39 weeks of gestation. It is worth highlighting that the rate of increase in the proportion of births at 39 weeks was higher than the rate of fall in caesarean sections between 2012 and 2019. The APC of gestational ages ≥42 weeks declined more sharply after the 2012 recommendation to terminate pregnancies upon reaching 41 completed weeks [24].

It is reasonable to assume that factors other than the implementation of women's health policies in the 1980s may have influenced the findings of the present study. In 2006, the women's organization *Rede Parto do Princípio* reported the abuse of caesarean delivery in the private health system to the Public Prosecutor's Office (MPF, acronym in Portuguese) [25,26]. The MPF accepted the complaint and filed a public interest civil lawsuit seeking to force the private health sector regulator, the *Agência Nacional de Saúde Suplementar* (ANS), to regulate the quality of obstetric services provided by the private health system [27], where caesarean section rates are particularly high, reaching 100% in some maternity units. On 6 January 2015, the ANS issued Normative Resolution 368 [28], which sets out measures to ensure that private health insurance policy-holders have access to data on caesarean section rates by healthcare operator and health facility and making the use of partographs and maternity notes mandatory. In March 2015, the ANS also launched the "Adequate Birth Project" [26] which promotes a set of strategies aimed at improving childbirth support to reduce caesarean section rates. In March 2016, the Federal Medical Council issued Resolution 2144 recommending that elective caesarean sections should only be performed from 39 weeks of gestation to guarantee the safety of the baby [29].

**Table 4. Annual percent change of live births according to gestational age by type of delivery. City of São Paulo, Brazil, 2012–2019.**

| Type of birth | 2012–2015 | | | | | | | 2016–2019 | | | | | | | 2012–2019 | | | | | | |
|---|---|---|---|---|---|---|---|---|---|---|---|---|---|---|---|---|---|---|---|---|---|
| | Vaginal | | | Cesarean | | | t-test | Vaginal | | | Cesarean | | | t-test | Vaginal | | | Cesarean | | | t-test |
| GA | rate (%) | | 95%CI | rate (%) | | 95%CI | 95%CI | rate (%) | | rate (%) | rate (%) | | rate (%) | 95%CI | rate (%) | | rate (%) | rate (%) | | rate (%) | 95%CI |
| < 34 | 0.2 | * | (0.2;0.2) | −3.2 | | (−10.0;4.2) | | −6.5 | * | (−7.1;-5.8) | 0.2 | * | (0.2;0.2) | ** | −10.9 | * | (−14.7;-6.9) | 1.9 | | (−0.9;4.7) | |
| 34 | −9.8 | | (−42.3;41.0) | 0.5 | | (−15.1;18.9) | | −10.3 | * | (−10.3;10.3) | 0 | * | (0.0;0.0) | ** | −9.4 | * | (−14.7;-3.8) | −0.9 | | (−3.6;1.8) | ** |
| 35 | −14.7 | * | (−25.8;-1.9) | −8.4 | * | (−12.3;-4.3) | ** | −8.4 | * | (−13.6;-2.8) | 2.8 | | (−8.6;15.6) | ** | −9.0 | * | (−12.4;-5.5) | −2.1 | | (−6.2;2.3) | ** |
| 36 | −13.9 | * | (−18.8;-8.7) | −6.2 | * | (−6.2;-6.2) | ** | −3.4 | * | (−3.4;-3.4) | 5.7 | * | (5.7;5.7) | ** | −7.1 | * | (−10.6;-3.5) | −2.9 | | (−7.1;1.4) | ** |
| 37 | −6.9 | * | (−7.6;-6.2) | −12.1 | * | (−21.2;-1.9) | ** | 0.0 | * | (0.0;0.0) | −0.2 | * | (−0.2;-0.2) | ** | −3.4 | * | (−6.0;-0.7) | −7.3 | * | (−12.7;-1.6) | ** |
| 38 | 1.4 | * | (1.4;1.4) | −4.1 | | (−15.3;8.7) | ** | −0.7 | * | (−0.7;-0.7) | −2.7 | * | (−2.7;-2.7) | ** | 1.2 | * | (0.6;1.7) | −7.5 | * | (−11.0;-3.9) | ** |
| 39 | 11.2 | * | (9.6;12.8) | 13.0 | * | (2.7;24.3) | | 2.8 | * | (2.8;2.8) | 3.0 | * | (1.5;4.6) | | 5.4 | * | (2.6;8.3) | 10.2 | * | (5.5;15.1) | ** |
| 40 | 7.9 | * | (5.5;10.3) | 10.7 | * | (0.6;21.7) | ** | 3.0 | * | (3.0;3.0) | −1.4 | | (−2.8;0.1) | ** | 4.2 | * | (2.5;5.9) | 6.4 | * | (1.3;11.8) | ** |
| 41 | −14.1 | * | (−14.1;-14.1) | −8.8 | * | (−15.9;-1.1) | ** | −9.8 | * | (−15.0;-4.4) | −3.8 | | (−9.3;2.0) | ** | −12.5 | * | (−14.4;-10.6) | −6.5 | * | (−8.5;-4.4) | ** |
| 42+ | −58.7 | * | (−59.9;-57.5) | −46.0 | * | (−48.4;-43.6) | ** | −28.9 | * | (−28.9;-28.9) | −21.7 | * | (−30.3;-11.9) | ** | −37.3 | * | (−48.2;-24.2) | −29.9 | * | (−38.1;-20.5) | |

*p < 0.05 rejects the null hypothesis.

**Statistically significant differences in percent change between vaginal and cesarean births (p < 0.05).

**Table 5. Annual percent change of live births according to gestational age by system (public or private). City of São Paulo, Brazil, 2012–2019.**

| System GA | 2012–2015 Public rate (%) | 95%CI | Private rate (%) | 95%CI | t-test | 2016–2019 Public rate (%) | 95%CI | Private rate (%) | 95%CI | t-test | 2012–2019 Public rate (%) | 95%CI | Private rate (%) | 95%CI | t-test |
|---|---|---|---|---|---|---|---|---|---|---|---|---|---|---|---|
| <34 | -12.7 * | (-15.2;-10.1) | -4.5 | (-10.6;2.0) | ** | -2.5 * | (-2.5;-2.5) | -3.4 | (-3.4;-3.4) |  | -4.5 * | (-8.6;-0.2) | -2.3 * | (-3.3;-1.2) | ** |
| 34 | -2.5 | (-39.9;58.1) | 0 | (0.0;0.0) |  | -7.7 * | (-7.7;-7.7) | -2.3 | (-2.3;-2.3) | ** | -4.7 | (-10.7;1.7) | -3.6 * | (-5.2;-2.0) |  |
| 35 | -11.7 | (-21.5;-0.7) | -10.5 * | (-12.4;-8.5) |  | -4.1 | (-14.0;7.1) | 1.4 | (-2.3;5.2) | ** | -5.2 * | (-9.2;-0.9) | -4.9 * | (-9.0;-0.7) |  |
| 36 | -11.1 * | (-14.9;-7.1) | -7.7 * | (-7.7;-7.7) | ** | 3.0 * | (3.0;3.0) | 0.2 | (-2.7;3.2) | ** | -3.6 * | (-7.7;0.7) | -5.6 * | (-9.6;-1.4) |  |
| 37 | -5.4 * | (-5.4;-5.4) | -14.1 * | (-24.2;-2.7) | ** | 2.3 * | (2.3;2.3) | -2.3 | (-5.8;1.4) | ** | -2.5 | (-5.6;0.7) | -8.4 * | (-13.7;-2.7) | ** |
| 38 | 1.2 * | (1.2;1.2) | -4.9 | (-17.3;9.3) | ** | 0.5 | (-1.0;1.9) | -4.9 | (-9.0;-0.7) | ** | 0.7 * | (0.1;1.2) | -7.3 * | (-10.8;-3.7) | ** |
| 39 | 11.9 * | (11.9;11.9) | 13.5 * | (0.2;28.6) |  | 3.0 * | (2.3;3.8) | 7.2 | (-0.4;15.3) |  | 5.4 * | (2.6;8.3) | 11.7 * | (5.8;17.9) | ** |
| 40 | 8.1 * | (7.4;8.9) | 13.2 | (-2.9;32.1) |  | 1.2 | (-0.3;2.7) | -1.8 | (-4.0;0.4) | ** | 4.0 * | (1.8;6.3) | 6.9 * | (0.7;13.5) | ** |
| 41 | -13.3 * | (-13.3;-13.3) | -1.4 | (-20.8;22.9) | ** | -13.5 * | (-13.5;13.5) | -2.7 | (-14.7;11.0) | ** | -12.9 * | (-14.3;-11.5) | -0.5 | (-4.2;3.4) | ** |
| 42 + | -57.0 * | (-57.0;-57.0) | -35.4 * | (-35.4;-35.4) | ** | -56.3 * | (-56.3;-56.3) | -21.1 | (-25.1;-17.0) | ** | -36.9 * | (-47.3;-24.5) | -24.7 * | (-30.2;-18.7) | ** |

*p<0.05 rejects the null hypothesis.

**Statistically significant differences in percent change between public and private services (p<0.05).

The combination of these factors helped draw attention to the excessive use of the caesarean section in the private sector, empowering women to make informed birth choices. Our findings show that 2015 was an inflection point for caesarean section rates in SP, with an increase in the proportion of this type of birth at 39 weeks up to the end of the time series. This rise was more pronounced in private sector births.

**Table 6. Annual percent change of live births according to gestational age by maternal age. City of São Paulo, Brazil, 2012–2019.**

| | 2012–2015 | | | | | | | | |
|---|---|---|---|---|---|---|---|---|---|
| Maternal age (years) | < 20 | | | 20–34 | | | 35 and over | | |
| GA | rate (%) | | IC$_{95\%}$ | rate (%) | | IC$_{95\%}$ | rate (%) | | IC$_{95\%}$ |
| < 34 | 0.2 | * | (0.2;0.2) | −8.4 | * | (−12.3;-4.3) | −15.3 | * | (−15.3;-15.3) |
| 34 | 0.5 | | (−39.0;65.4) | −4.5 | | (−24.5;20.7) | −3.4 | | (−27.9;29.5) |
| 35 | −11.1 | * | (−18.0;-3.6) | −10.3 | * | (−18.4;-1.3) | −15.7 | * | (−15.7;-15.7) |
| 36 | −7.7 | * | (−11.7;-3.6) | −9.2 | * | (−9.2;-9.2) | −11.5 | * | (−17.7;-4.8) |
| 37 | −7.3 | * | (−10.7;-3.9) | −11.5 | * | (−18.3;-4.1) | −11.9 | * | (−19.3;-3.8) |
| 38 | −0.2 | | (−3.1;2.7) | −4.3 | | (−13.0;5.3) | −0.2 | | (−13.2;14.7) |
| 39 | 11.9 | * | (11.9;11.9) | 12.5 | * | (7.6;17.5) | 13.0 | * | (2.0;25.2) |
| 40 | 9.4 | * | (7.0;11.8) | 7.9 | * | (7.9;7.9) | 10.2 | | (−3.5;25.7) |
| 41 | −15.3 | * | (−15.3;-15.3) | −11.3 | * | (−12.6;-10.0) | −9.0 | | (−16.7;-0.6) |
| 42 and over | −55.6 | * | (−62.8;-47.1) | −52.0 | * | (−55.1;-4.8) | −50.0 | * | (−62.4;-33.5) |
| | 2016–2019 | | | | | | | | |
| Maternal age (years) | < 20 | | | 20–34 | | | 35 and over | | |
| GA | rate (%) | | IC$_{95\%}$ | rate (%) | | IC$_{95\%}$ | rate (%) | | IC$_{95\%}$ |
| < 34 | −1.6 | | (−7.9;5.1) | −4.9 | * | (−4.9;-4.9) | 0.2 | * | (0.2;0.2) |
| 34 | 0.0 | * | (0.0;0.0) | −8.6 | * | (−11.9;-5.2) | 0 | * | (0.0;0.0) |
| 35 | −7.5 | * | (−7.5;-7.5) | −2.5 | | (−11.4;7.2) | −1.1 | | (−14.6;14.5) |
| 36 | −6.0 | | (−11.4;0.0) | 0.9 | * | (0.9;0.9) | 5.0 | | (0.0;9.7) |
| 37 | −5.2 | * | (−5.2;-5.2) | −1.1 | * | (−1.1;-1.1) | −0.7 | * | (−0.7;-0.7) |
| 38 | −0.5 | | (−5.4;4.8) | −3.4 | * | (−4.1;-2.7) | −2.1 | * | (−2.1;-2.1) |
| 39 | 6.4 | * | (6.4;6.4) | 3.5 | * | (2.8;4.3) | 0.5 | * | (0.5;0.5) |
| 40 | 1.6 | * | (1.6;1.6) | 2.6 | * | (1.8;3.3) | 3.5 | * | (1.3;5.8) |
| 41 | −5.4 | * | (−6.8;-4.0) | −8.2 | * | (−8.2;-8.2) | −7.1 | * | (−7.1;-7.1) |
| 42 and over | −27.4 | * | (−27.4;-27.4) | −23.6 | * | (−23.6;-23.6) | −27.9 | * | (−37.3;-17.1) |
| | 2012–2019 | | | | | | | | |
| Maternal age (years) | < 20 | | | 20–34 | | | 35 and over | | |
| GA | rate (%) | | IC$_{95\%}$ | rate (%) | | IC$_{95\%}$ | rate (%) | | IC$_{95\%}$ |
| < 34 | −4.5 | * | (−8.6;-0.2) | −3.6 | * | (−5.7;-1.5) | −2.3 | | (−5.9;1.5) |
| 34 | −5.8 | | (−13.2;2.2) | −5.2 | * | (−8.2;-2.0) | −3.2 | | (−8.8;2.8) |
| 35 | −7.3 | * | (−9.3;-5.3) | −5.2 | * | (−8.7;-1.5) | −5.4 | | (−10.4;-0.0) |
| 36 | −4.7 | * | (−6.8;-2.6) | −5.8 | * | (−9.3;-2.2) | −4.7 | | (−9.8;0.6) |
| 37 | −5.4 | * | (−6.4;-4.3) | −7.3 | * | (−11.7;-2.7) | −6.9 | * | (−11.8;-1.7) |
| 38 | 0.7 | | (−0.4;1.8) | −6.2 | * | (−8.8;-3.7) | −5.6 | * | (−9.1;-1.9) |
| 39 | 5.4 | * | (3.2;7.8) | 8.1 | * | (5.2;11.1) | 9.1 | * | (3.9;14.6) |
| 40 | 4.5 | * | (1.7;7.4) | 7.2 | * | (4.3;10.1) | 7.9 | * | (4.4;11.5) |
| 41 | −12.3 | * | (−15.1;-9.4) | −7.5 | * | (−9.0;-6.0) | −5.2 | * | (−7.7;-2.5) |
| 42 and over | −35.9 | * | (−45.2;-24.9) | −32.1 | * | (−42.3;-20.0) | −32.5 | * | (−40.8;-23.1) |

*p<0.05 rejects the null hypothesis.

**Table 7. Annual percent change of live births according to gestational age by education level. City of São Paulo. 2012–2019.**

| | 2012–2015 | | | | | | | | |
|---|---|---|---|---|---|---|---|---|---|
| Education | Did not complete elementary school/ Did not complete elementary school | | | Completed high school | | | Completed or undergoing higher education | | |
| GA | rate (%) | | IC95% | rate (%) | | IC95% | rate (%) | | IC95% |
| < 34 | 0.2 | * | (0.2;0.2) | 0.2 | * | (0.2;0.2) | −2.5 | | (−12.7;8.8) |
| 34 | −3.2 | | (−45.3;71.5) | −2.7 | | (−26.4;28.5) | 0 | | (0.0;0.0) |
| 35 | −13.9 | * | (−15.2;-12.6) | −9.8 | | (−21.6;3.6) | −11.7 | * | (−13.6;-9.7) |
| 36 | −12.9 | * | (−16.7;-9.0) | −8.2 | * | (−8.2;-8.2) | −8.0 | * | (−8.0;-8.0) |
| 37 | −5.8 | * | (−9.2;-2.3) | −10.1 | * | (−16.4;-3.2) | −16.1 | * | (−23.1;-8.3) |
| 38 | 0.9 | * | (0.9;0.9) | −2.7 | | (−8.3;3.1) | −7.7 | | (−20.3;6.8) |
| 39 | 0.5 | * | (0.5;0.5) | 11.9 | * | (8.7;15.3) | 13.5 | | (0.2;28.6) |
| 40 | 0.5 | * | (0.5;0.5) | 8.4 | * | (8.4;8.4) | 21.6 | * | (5.0;40.8) |
| 41 | 0.2 | * | (0.2;0.2) | −12.3 | * | (−12.3;-12.3) | 4.5 | * | (4.5;4.5) |
| 42 and over | −55.3 | * | (−55.3;-55.3) | −52.4 | * | (−54.4;-50.2) | −33.8 | * | (−47,6;-16.3) |
| | 2016 a 2019 | | | | | | | | |
| Education | Did not complete elementary school/ Did not complete elementary school | | | Completed high school | | | Completed or undergoing higher education | | |
| GA | rate (%) | | IC95% | rate (%) | | IC95% | rate (%) | | IC95% |
| < 34 | 0 | | (0.0;0.0) | −2.5 | | (−4.6;-0.3) | −3.2 | | (−8.7;2.7) |
| 34 | −1.8 | | (−17.7;17.1) | −6.7 | | (−13.3;0.4) | 0 | | (0.0;0.0) |
| 35 | −0.2 | | (−12.6;13.8) | −2.3 | | (−8.5;4,4) | −1.8 | | (−12.0;9.6) |
| 36 | 6.9 | * | (6.9;6.9) | −1.4 | * | (−2.1;-0,6) | 3.0 | * | (3.0;3.0) |
| 37 | 3.0 | | (−3.5;10.1) | −1.4 | * | (−1.4;-1.4) | −0.2 | * | (−0.2;-0.2) |
| 38 | 0.5 | * | (0.5;0.5) | −2.1 | | (−4.2;0.1) | −4.3 | * | (−4.3;-4.3) |
| 39 | 4.2 | * | (4.2;4.2) | 3.3 | * | (2.5;4.0) | 2.1 | * | (2.1;2.1) |
| 40 | −0.5 | * | (−0.5;-0.5) | 2.8 | * | (2.8;2.8) | 4.0 | * | (3.2;4.8) |
| 41 | −16.4 | * | (−17.0;-15.8) | −9.4 | * | (−9.4;-9.4) | 5.9 | | (−0.1;12.3) |
| 42 and over | −29.0 | * | (−29.0;-29.0) | −26.9 | * | (−26.9;-26.9) | −15.1 | * | (−15.1;-15.1) |
| | 2012–2019 | | | | | | | | |
| Education | Did not complete elementary school/ Did not complete elementary school | | | Completed high school | | | Completed or undergoing higher education | | |
| GA | rate (%) | | IC95% | rate (%) | | IC95% | rate (%) | | IC95% |
| < 34 | −3.8 | | (−8.9;1.5) | −3.4 | | (−7.0;0.4) | −1.8 | | (−3.9;0.3) |
| 34 | −6.0 | | (−13.4;2.0) | −3.8 | | (−7.9;0.4) | −3.8 | * | (−5.4;-2.3) |
| 35 | −3.6 | | (−9.2;2,3) | −4.1 | * | (−7.6;-0.3) | −6.9 | * | (−10.4;-3.3) |
| 36 | −2.7 | | (−7.9;2.7) | −4.9 | * | (−8.5;-1.2) | −6.0 | * | (−10.0;-1.8) |
| 37 | −2.3 | | (−5.9;1.5) | −6.0 | * | (−9.5;-2.4) | −9.6 | * | (−15.4;-3.5) |
| 38 | 0.7 | * | (0.7;0.7) | −4.7 | * | (−6.3;-3.2) | −9.4 | * | (−13.3;-5.4) |
| 39 | 5.7 | * | (2.8;8.6) | 7.2 | * | (4.3;10.1) | 10.4 | * | (4.6;16.6) |
| 40 | 4.0 | * | (0.1;8.0) | 5.7 | * | (3.4;8.0) | 14.3 | * | (7,1;22,0) |
| 41 | −14.1 | * | (−15.0;-13.2) | −9.6 | * | (−11.1;-8.1) | 9.6 | * | (5.5;13.9) |
| 42 and over | −36.9 | * | (−46.1;-26.1) | −33.3 | * | (−42.7;-22.3) | −18.5 | * | (−26.1;-10.1) |

*p<0.05 rejects the null hypothesis.

Early caesarean sections without a medical reason can have adverse effects on short- and long-term maternal and infant health and well-being. At the end of pregnancy, every day counts and the more the birth is brought forward the higher the risk of infant mortality [30]. Thus, monitoring trends in GA at birth can help assess not only chances of survival but also future health. The reduction in preterm and early term births and increase in the proportion of births at later GAs is therefore good news. Nevertheless, induced labor and caesarean births remain the most common mode of delivery in SP, showing that non-evidenced based practices prevail in the country's most populous city.

## Limitations

This study has inherent limitations associated with the use of secondary databases. While these sources are characterized by high quality and reliability, they do not allow for an in-depth exploration of childbirth humanization or an assessment of maternal satisfaction with the care received. Nonetheless, the findings—particularly the observed reduction in preterm and early-term births among cesarean deliveries—indicate that policies aimed at improving maternal and postpartum care have already yielded positive outcomes, underscoring the importance of their continuation and refinement.

Additionally, the scope of social variables analyzed was limited to maternal education and age. Future research could expand this framework by incorporating other relevant factors, such as the mother's residential neighborhood and racial background, to provide a more comprehensive understanding of childbirth care disparities.

## Recommendations

There is still a long way to go to improve intrapartum care in Brazil, which entails the uptake of national and international recommendations [31–33], including:

(1) the use of evidence-based practices, promoting a change in culture, where childbirth is not seen primarily as a high-tech medical event.

(2) the dissemination of an antenatal intrapartum care model involving interdisciplinary teams, including autonomous obstetric nurses and midwives for low-risk pregnancies and the active participation of women in their childbirth [34].

(3) the strengthening of the country's public health system, the *Sistema Único de Saúde* (SUS) or Unified Health System, and its principles and guidelines, including public participation and the dissemination of information on women's rights.

(4) the transparent monitoring of data and service and professional performance indicators.

(5) the regulation of intrapartum care practices and price setting in the private sector to ensure good ethical standards.

It is hoped that the effective implementation of these recommendations will make birth safer and improve women's satisfaction with the childbirth experience, positively influencing long-term infant outcomes. This constitutes a broad health care agenda that merits priority attention if we want to promote the health of women and the next generations.

## Acknowledgments

This study is part of the **"Potential pregnancy days lost" (PPDL**) **project**, which provides an innovative measure of gestational age to assess maternal and child health interventions and outcomes. The project was developed in partnership with the São Paulo City Department of Health, Bill & Melinda Gates Foundation (reference number INV-027961), and National Council for Scientific and Technological Development (CNPq/DECIT-MS reference number 445116/2020–0).

## Author contributions

**Conceptualization:** Margarida Maria Tenório de Azevedo Lira, Marina de Freitas, Eliana de Aquino Bonilha, Célia Maria Castex Aly, Patrícia Carla dos Santos, Carmen Simone Grilo Diniz.

**Data curation:** Margarida Maria Tenório de Azevedo Lira, Eliana de Aquino Bonilha, Célia Maria Castex Aly, Patrícia Carla dos Santos, Carmen Simone Grilo Diniz.

**Formal analysis:** Margarida Maria Tenório de Azevedo Lira, Marina de Freitas, Eliana de Aquino Bonilha, Célia Maria Castex Aly, Patrícia Carla dos Santos, Denise Yoshie Niy, Carmen Simone Grilo Diniz.

**Funding acquisition:** Eliana de Aquino Bonilha, Carmen Simone Grilo Diniz.

**Investigation:** Margarida Maria Tenório de Azevedo Lira, Marina de Freitas, Eliana de Aquino Bonilha, Célia Maria Castex Aly, Denise Yoshie Niy, Carmen Simone Grilo Diniz.

**Methodology:** Margarida Maria Tenório de Azevedo Lira, Marina de Freitas, Eliana de Aquino Bonilha, Célia Maria Castex Aly, Patrícia Carla dos Santos, Carmen Simone Grilo Diniz.

**Project administration:** Eliana de Aquino Bonilha, Carmen Simone Grilo Diniz.

**Resources:** Eliana de Aquino Bonilha, Carmen Simone Grilo Diniz.

**Software:** Margarida Maria Tenório de Azevedo Lira.

**Supervision:** Margarida Maria Tenório de Azevedo Lira, Marina de Freitas, Eliana de Aquino Bonilha, Célia Maria Castex Aly, Patrícia Carla dos Santos, Carmen Simone Grilo Diniz.

**Validation:** Margarida Maria Tenório de Azevedo Lira, Marina de Freitas, Eliana de Aquino Bonilha, Célia Maria Castex Aly, Patrícia Carla dos Santos, Denise Yoshie Niy, Carmen Simone Grilo Diniz.

**Visualization:** Margarida Maria Tenório de Azevedo Lira, Marina de Freitas, Eliana de Aquino Bonilha, Célia Maria Castex Aly, Patrícia Carla dos Santos, Carmen Simone Grilo Diniz.

**Writing – original draft:** Margarida Maria Tenório de Azevedo Lira, Marina de Freitas, Eliana de Aquino Bonilha, Célia Maria Castex Aly, Patrícia Carla dos Santos, Denise Yoshie Niy, Carmen Simone Grilo Diniz.

**Writing – review & editing:** Margarida Maria Tenório de Azevedo Lira, Marina de Freitas, Eliana de Aquino Bonilha, Célia Maria Castex Aly, Patrícia Carla dos Santos, Denise Yoshie Niy, Carmen Simone Grilo Diniz.

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
