## [Decision Letter · Decision Letter 0]

14 Mar 2025

PONE-D-25-03233Trends in gestational age at birth in the city of São Paulo, Brazil between 2012 and 2019PLOS ONE

Dear Dr. Niy,

Thank you for submitting your manuscript to PLOS ONE. After careful consideration, we feel that it has merit but does not fully meet PLOS ONE’s publication criteria as it currently stands. Therefore, we invite you to submit a revised version of the manuscript that addresses the points raised during the review process.

We look forward to receiving your revised manuscript.

Kind regards,

André Ricardo Ribas Freitas

Academic Editor

PLOS ONE

Journal Requirements:

Reviewers' comments:

Reviewer's Responses to Questions

**Comments to the Author**

1. Is the manuscript technically sound, and do the data support the conclusions?

Reviewer #1: Partly

Reviewer #2: Yes

2. Has the statistical analysis been performed appropriately and rigorously? 

Reviewer #1: No

Reviewer #2: Yes

3. Have the authors made all data underlying the findings in their manuscript fully available?

Reviewer #1: Yes

Reviewer #2: Yes

4. Is the manuscript presented in an intelligible fashion and written in standard English?

Reviewer #1: No

Reviewer #2: Yes

5. Review Comments to the Author

Reviewer #1: Type of study and data source

It would be interesting to put the website address from which the data was downloaded (line 81).

I think the study is an observational follow-up study and not an ecological one.

Study population

In lines 86 and 87 the authors say: “Births without information on type of delivery, GA, and type of pregnancy were excluded (Table 1)”. Did all live births in SP between 2012 and 2019 have their gestacional age and birth weight recorded?

In the version of the article I had access to,I coudn´t find Figure 1 that the authors mention in line 98. On line 100 there is only the title of the figure: “Figure 1. Live births by type of pregnancy. City of São Paulo, Brazil, 2012-2019. ” Please check.

Statistical analysis

In lines 110, 111 and 112 the text is in Portuguese. Please check.

Results

In Table 2, the decimal separator for the percentages has a comma, but the manuscript is in English. Please check.

The sum of the percentages of maternal age and education (Table 2) does not add up to 100%. Please check. Why don´t the percentages of Labor induction and CS before labor add up to 100%?

Again, I don´t have any of the figures mentioned in the article, so I couldn´t follow the comments the authors make between lines 139 and 167.

In Table 3, the decimal separator for the percentages has a comma, but the manuscript is in English. Please check.

In Table 3 I suggest that instead of rate the authors use APC (average annual change) in percent. The title of the table reads “Average anual percent change in proportion”. Percent in proportion, what does that mean?

In Tables 4 and 5 keepthe format of the presentation of the columns as in Table 3, ie, first the columns 2012 to 2015, then 2016 to 2019 and the last columns for 2012 to 2019. Was the t-test that the authors show obtained from the regression model or were there seral t-tests comparing the APC’s? In the methodology, the authors make no mention of this test and don´t even say anything about the level os sigificance adopted in the work.

In Table 6, the authors didn´t analyze the two periods as they did in the other tables, why?

In Tables 4 to 6 the APC for GA 42 and over are very high when compared to the other GA’s, but the authors make no comment on this. Why?

References

Reference 19 is missing the year of publication.

Rêgo M, Leão DC, Luiza M, Riesco G, Schneck CA, Angelo M. Reflexões sobre o excesso de cesarianas no Brasil e a autonomia das mulheres Reflections on the excessive rates of cesareans in Brazil and the empowerment of women. : 2395–2400.

In reference 24, the final part of the article´s title is missing

Queiroz MR, Junqueira MER, Lay AAR, Bonilha EDA, Borba MF, Aly CMC, et al. Neonatal mortality by gestational age in days in infants born at term : A cohort study in Sao. PLoS ONE. 2022;17: 1–11. doi:10.1371/journal.pone.0277833

Reviewer #2: The aim of this study was to analyze trends in GA at birth and the contribution of associated factors in the city of São Paulo during the period 2012-2019, through an ecological time-series study, using data from Brazil’s national live births information system (SINASC). The topic is relevant to Public Health. However, the manuscript needs a set of adjustments.

Introduction

- I suggest inserting the paragraph about SINASC (lines 59 to 63) in the Materials and Methods section.

Materials and Methods section

- the objective of the study is to analyze the trends in Gestational Age at birth in the city of São Paulo. However, when considering births that occurred in the city of São Paulo, the total number of live births in the city of São Paulo is obtained, regardless of the mother’s place of residence. How can this fact affect the results and conclusions of the study? Therefore, I suggest two possibilities: to present the answer to this question in the limitations of the Discussion section, or to redo the analyses considering only live births of mothers residing in the city of São Paulo who gave birth in the city itself.

- In Table 1, correct the excluded values of the variables “Gestational Age”, “birth weight” and “out of hospital births”, adding the values without information.

- I suggest adding a topic to describe the variables used in the study.

- Why was the variable “race/skin color of the mother”, available in SINASC, not included in the Prais-Winsten regression analysis? I suggest justifying or adding this variable.

- I suggest standardizing the use of the term “Annual Percent Change (APC)” instead of “Percent change rate” in the Statistical analysis section.

- In lines 110-112, translate the sentence into English.

- Correct the formula in line 113, replacing “e” with “10”.

- Cite the version of the SPSS software.

- Cite and justify the application of the “t-test” in tables 3 and 4. Why was this statistical test not presented in table 6?

- Since ensuring quality information is an essential condition for the analysis of epidemiological indicators (proportions), I suggest calculating the completeness of the variables used in the study, in the period 2012-2019.

Results

- In tables 3, 4, 5, 6, I suggest "APC" instead of "rate" and adding the respective p values.

- In lines 172-173, the authors report that “These reductions were more pronounced in the first four-year period (2012-2015)”. I suggest applying a statistical test to identify differences between the periods 2012-2015” and “2016-2019”.

- In line 183, correct the APC value “for vaginal births”.

- In lines 208-209, the authors report that “Rates were higher among mothers aged 35 years and over and who had completed more than 12 years of education”. I suggest applying a statistical test to identify differences between age groups and educational levels.

- Why in Table 6 were the same analyses not performed for the periods “2012-2015” and “2016-2019”? I suggest standardization according to the results presented in Tables 3, 4 and 5.

- For Figures 2, 3, 4 and 5, I suggest applying a statistical test to identify differences in the proportions of the variables during the study period.

Discussion

- The discussion section could be more in-depth, citing other studies with similar designs (ecological time-series study) that evaluated the same outcome (trends in GA at birth) in Brazil and other countries.

- Included the limitations of ecological studies and other limitations of the analysis of Health Information Systems.

I suggest including a paragraph in the discussion addressing how social inequalities in the city of São Paulo and the COVID-19 pandemic can affect the conclusions of the study.

6. PLOS authors have the option to publish the peer review history of their article (what does this mean? ). If published, this will include your full peer review and any attached files.

**Do you want your identity to be public for this peer review?** For information about this choice, including consent withdrawal, please see our Privacy Policy .

Reviewer #1: No

Reviewer #2: No

---

## [Author Response · Author response to Decision Letter 1]

12 May 2025

Dear Editor,

We are thankful for the thorough review of our manuscript. We have made our best to meet the reviewers’ suggestions, as described below.

Journal Requirements:

We did our best to meet the journal's style requirements.

We included the requested information in the Methods section, as follows:

“This study is part of the Potential Pregnancy Days Lost project, which was approved by the Research Ethics Committee of the University of São Paulo's School of Public Health (CAAE: 98163018.2.0000.5421), on October 11, 2018. Since the analysis used secondary data, individual consent was not required.”

3. We note that you have included the phrase “data not shown” in your manuscript. Unfortunately, this does not meet our data sharing requirements. PLOS does not permit references to inaccessible data. We require that authors provide all relevant data within the paper, Supporting Information files, or in an acceptable, public repository.

All data can be accessed in Harvard Dataverse, and we included a reference for that (https://dataverse.harvard.edu/dataverse/DPGP).

4. Please review your reference list to ensure that it is complete and correct. If you have cited papers that have been retracted, please include the rationale for doing so in the manuscript text, or remove these references and replace them with relevant current references.

We reviewed the reference list and to the best of our knowledge no cited paper has been retracted. We added some references, as required and explained below.

Reviewer #1:

Type of study and data source

1. It would be interesting to put the website address from which the data was downloaded (line 81).

The databases were provided by the Health Department via ftp link. To clarify that, we included the following information:

“The anonymized databases were provided by the Sao Paulo Municipal Health Department on May 6, 2020.”

2. I think the study is an observational follow-up study and not an ecological one.

We changed the description to "observational time-series study", as required.

Study population

3. In lines 86 and 87 the authors say: “Births without information on type of delivery, GA, and type of pregnancy were excluded (Table 1)”. Did all live births in SP between 2012 and 2019 have their gestational age and birth weight recorded?

No. The proportion of unavailable information for gestational age ranged from 0.01% to 1.20%, with an average of 0.25% (2012-2019). For birth weight, the proportion of unavailable data varied between 0.001% and 0.007%, with an average of 0.003% during the period.

We excluded all live births that did not have one or more of the following information: maternal age, type of pregnancy, gestational age, birth weight, place of birth or type of delivery. We modified the above mentioned paragraph to clarify that, and included the following information:

4. In the version of the article I had access to,I couldn't find Figure 1 that the authors mention in line 98. On line 100 there is only the title of the figure: “Figure 1. Live births by type of pregnancy. City of São Paulo, Brazil, 2012-2019. ” Please check.

We apologize for that - we uploaded all the figures again, following the journal's requirements. Additionally, we uploaded a pdf version with all figures.

Statistical analysis

5. In lines 110, 111 and 112 the text is in Portuguese. Please check.

We apologize for that. The text was translated to English.

Results

6. In Table 2, the decimal separator for the percentages has a comma, but the manuscript is in English. Please check.

We checked all the tables so that all values have a dot as a decimal separator.

7. The sum of the percentages of maternal age and education (Table 2) does not add up to 100%. Please check. Why don't the percentages of Labor induction and CS before labor add up to 100%?

We revised the table to clarify how the calculations were performed and we presented all the values related to the mentioned variables.

8. Again, I don't have any of the figures mentioned in the article, so I couldn´t follow the comments the authors make between lines 139 and 167.

We apologize for that - we uploaded all the figures, following the journal's requirements.

9. In Table 3, the decimal separator for the percentages has a comma, but the manuscript is in English. Please check.

We checked all the tables so that all values have a dot as a decimal separator.

10. In Table 3 I suggest that instead of rate the authors use APC (average annual change) in percent. The title of the table reads “Average annual percent change in proportion”. Percent in proportion, what does that mean?

We changed the text and the table's title, as requested.

11. In Tables 4 and 5 keep the format of the presentation of the columns as in Table 3, ie, first the columns 2012 to 2015, then 2016 to 2019 and the last columns for 2012 to 2019.

We reworked the tables, as requested.

12. Was the t-test that the authors show obtained from the regression model or were there seral t-tests comparing the APC’s? In the methodology, the authors make no mention of this test and don't even say anything about the level of significance adopted in the work.

The test had been done. We updated the methods section to include this information, as requested.

13. In Table 6, the authors did not analyze the two periods as they did in the other tables, why?

We reworked the table, as suggested.

14. In Tables 4 to 6 the APC for GA 42 and over are very high when compared to the other GA’s, but the authors make no comment on this. Why?

We added comments about this finding in the Results section.

The APC of gestational ages ≥42 weeks declined more sharply after the 2012 recommendation to terminate pregnancies upon reaching 41 completed weeks (reference included: Gülmezoglu, A. Metin, et al. "Induction of labour for improving birth outcomes for women at or beyond term." Cochrane database of systematic reviews 6 (2012).

References

15. Reference 19 is missing the year of publication.

We included the year of publication.

16. In reference 24, the final part of the article’s title is missing.

We included the missing information.

Queiroz MR, Ramos Junqueira ME, Roman Lay AA, Bonilha EA, Borba MF, Castex Aly CM, Moreira RA, Diniz CSG. Neonatal mortality by gestational age in days in infants born at term: A cohort study in Sao Paulo city, Brazil. PLoS One. 2022 Nov 21;17(11):e0277833. doi: 10.1371/journal.pone.0277833. PMID: 36409732; PMCID: PMC9678289.

Reviewer #2:

1. I suggest inserting the paragraph about SINASC (lines 59 to 63) in the Materials and Methods section.

We added it in the Methods section: The Live Births Information System of Brazil (SINASC) was established in 1996 to systematically collect data on live births across the national territory. Designed to support all levels of Brazil’s healthcare system, SINASC has consistently demonstrated high coverage, completeness, and reliability in its recorded variables. This ensures its capacity to fulfill its primary objective: to provide comprehensive and objective analyses of the healthcare landscape, thereby informing policies that enhance maternal and child health.

In São Paulo, efforts to improve data quality have included rigorous monitoring and continuous professional training for those responsible for completing and inputting information into the Certificate of Live Birth (CLB), SINASC’s foundational reporting form. The training process encompasses the development of educational materials, seminars, and both individual and group workshops. Additionally, healthcare facilities that conduct births and adhere to established standards of completeness and timely data entry receive annual certification through the "SINASC Seal," reinforcing quality assurance and data integrity.

Materials and Methods section

2. The objective of the study is to analyze the trends in Gestational Age at birth in the city of São Paulo. However, when considering births that occurred in the city of São Paulo, the total number of live births in the city of São Paulo is obtained, regardless of the mother’s place of residence. How can this fact affect the results and conclusions of the study? Therefore, I suggest two possibilities: to present the answer to this question in the limitations of the Discussion section, or to redo the analyses considering only live births of mothers residing in the city of São Paulo who gave birth in the city itself.

We analyzed births occurring in São Paulo city to assess municipal childbirth care provision, providing actionable data for health system managers. We added this information to the Methods section.

3. In Table 1, correct the excluded values of the variables “Gestational Age”, “birth weight” and “out of hospital births”, adding the values without information.

We added information on variables’ completeness to the Methods section.

4. I suggest adding a topic to describe the variables used in the study.

We added the description of the studied variables to the Methods section, as suggested.

5. Why was the variable “race/skin color of the mother”, available in SINASC, not included in the Prais-Winsten regression analysis? I suggest justifying or adding this variable.

The socioeconomic variable chosen for analysis was “maternal education". In future analysis we will consider other relevant variables, such as “race/skin color of the mother”.

6. I suggest standardizing the use of the term “Annual Percent Change (APC)” instead of “Percent change rate” in the Statistical analysis section.

We adopted the term “annual percent change", as suggested.

7. In lines 110-112, translate the sentence into English.

We apologize for that. The text was translated to English.

8. Correct the formula in line 113, replacing “e” with “10”.

We corrected the formula, as required.

9. Cite the version of the SPSS software.

We added the required information to the Methods section.

10. Cite and justify the application of the “t-test” in tables 3 and 4. Why was this statistical test not presented in table 6?

The t-test (for independent proportions) was used to assess differences in percentage change rates between the public and private sectors and the type of birth. We added this information to the Methods section.

11. Since ensuring quality information is an essential condition for the analysis of epidemiological indicators (proportions), I suggest calculating the completeness of the variables used in the study, in the period 2012-2019.

We added the information about the variables’ completeness to the Methods section, as requested. From 2012 to 2019, the variables’ completeness ranged from 98.9% to 100.0%.

Results

12. In tables 3, 4, 5, 6, I suggest "APC" instead of "rate" and adding the respective p values.

We reworked the tables, as suggested.

13. In lines 172-173, the authors report that “These reductions were more pronounced in the first four-year period (2012-2015)”. I suggest applying a statistical test to identify differences between the periods 2012-2015” and “2016-2019”.

Resposta

14. In line 183, correct the APC value “for vaginal births” .

We corrected the value.

In the period 2012-2015, the annual percent change in the proportion of caesarean births at 38 weeks was −4.1%, compared to 1.4% for vaginal births. This difference was statistically significant. For 39 and 40 weeks, there was a higher percentage increase in cesarean sections compared to vaginal deliveries, but this difference was not significant. The proportion of preterm births decreased across all gestational ages under 37 weeks. This reduction was greater in vaginal births.

15. In lines 208-209, the authors report that “Rates were higher among mothers aged 35 years and over and who had completed more than 12 years of education”. I suggest applying a statistical test to identify differences between age groups and educational levels.

We reworked the analysis to better describe the comparisons made.

16.- Why in Table 6 were the same analyses not performed for the periods “2012-2015” and “2016-2019”? I suggest standardization according to the results presented in Tables 3, 4 and 5.

We reworked all the tables, as suggested.

17. For Figures 2, 3, 4 and 5, I suggest applying a statistical test to identify differences in the proportions of the variables during the study period.

The figures were calculated based on the proportion of births occurring in each gestational week, illustrating the annual distribution of gestational age (GA) during the study period. After descriptive analysis, we performed statistical testing to assess trends and associated factors.

Discussion

18. The discussion section could be more in-depth, citing other studies with similar designs (ecological time-series study) that evaluated the same outcome (trends in GA at birth) in Brazil and other countries.

We added the required information and a new reference (AMYX et al., 2024), as suggested.

19. Included the limitations of ecological studies and other limitations of the analysis of Health Information Systems.

We added the required information.

20. I suggest including a paragraph in the discussion addressing how social inequalities in the city of São Paulo and the COVID-19 pandemic can affect the conclusions of the study.

The study period did not include the COVID-19 pandemic.

---

## [Decision Letter · Decision Letter 1]

20 Aug 2025

Trends in gestational age at birth in the city of São Paulo, Brazil between 2012 and 2019

PONE-D-25-03233R1

Dear Dr. Niy,

We’re pleased to inform you that your manuscript has been judged scientifically suitable for publication and will be formally accepted for publication once it meets all outstanding technical requirements.

Kind regards,

Leonardo António Chavane, M.D., MPH, PhD

Academic Editor

PLOS ONE

Additional Editor Comments (optional):

Reviewers' comments:

Reviewer's Responses to Questions

**Comments to the Author**

1. If the authors have adequately addressed your comments raised in a previous round of review and you feel that this manuscript is now acceptable for publication, you may indicate that here to bypass the “Comments to the Author” section, enter your conflict of interest statement in the “Confidential to Editor” section, and submit your "Accept" recommendation.

Reviewer #2: (No Response)

Reviewer #3: All comments have been addressed

2. Is the manuscript technically sound, and do the data support the conclusions?

Reviewer #2: Yes

Reviewer #3: Yes

3. Has the statistical analysis been performed appropriately and rigorously? 

Reviewer #2: Yes

Reviewer #3: Yes

4. Have the authors made all data underlying the findings in their manuscript fully available?

Reviewer #2: Yes

Reviewer #3: Yes

5. Is the manuscript presented in an intelligible fashion and written in standard English?

Reviewer #2: Yes

Reviewer #3: Yes

6. Review Comments to the Author

Reviewer #2: In line 200, correct the APC value for the period 2012 to 2015 according to Table 3. The APC was -52.8%. Please verify.

On line 209, correct the APC for vaginal births to -3.4%. Please check.

Reviewer #3: (No Response)

7. PLOS authors have the option to publish the peer review history of their article (what does this mean? ). If published, this will include your full peer review and any attached files.

**Do you want your identity to be public for this peer review?** For information about this choice, including consent withdrawal, please see our Privacy Policy .

Reviewer #2: No

Reviewer #3: No

---

## [Editor Report · Acceptance letter]

PONE-D-25-03233R1

PLOS ONE

Dear Dr. Niy,

I'm pleased to inform you that your manuscript has been deemed suitable for publication in PLOS ONE. Congratulations! Your manuscript is now being handed over to our production team.

Kind regards,

on behalf of

Dr. Leonardo António Chavane

Academic Editor

PLOS ONE